# Biomarkers, Master Regulators and Genomic Fabric Remodeling in a Case of Papillary Thyroid Carcinoma

**DOI:** 10.3390/genes11091030

**Published:** 2020-09-02

**Authors:** Dumitru A. Iacobas

**Affiliations:** Personalized Genomics Laboratory, CRI Center for Computational Systems Biology, Roy G Perry College of Engineering, Prairie View A&M University, Prairie View, TX 77446, USA; daiacobas@pvamu.edu; Tel.: +1-936-261-9926

**Keywords:** 8505C cell line, apoptosis, BCPAP cell line, BRAF, CFLAR, IL6, oxidative phosphorylation, SPINT2, thyroid hormone synthesis, weighted pathway regulation

## Abstract

Publicly available (own) transcriptomic data have been analyzed to quantify the alteration in functional pathways in thyroid cancer, establish the gene hierarchy, identify potential gene targets and predict the effects of their manipulation. The expression data have been generated by profiling one case of papillary thyroid carcinoma (PTC) and genetically manipulated BCPAP (papillary) and 8505C (anaplastic) human thyroid cancer cell lines. The study used the genomic fabric paradigm that considers the transcriptome as a multi-dimensional mathematical object based on the three independent characteristics that can be derived for each gene from the expression data. We found remarkable remodeling of the thyroid hormone synthesis, cell cycle, oxidative phosphorylation and apoptosis pathways. Serine peptidase inhibitor, Kunitz type, 2 (*SPINT2*) was identified as the Gene Master Regulator of the investigated PTC. The substantial increase in the expression synergism of *SPINT2* with apoptosis genes in the cancer nodule with respect to the surrounding normal tissue (NOR) suggests that *SPINT2* experimental overexpression may force the PTC cells into apoptosis with a negligible effect on the NOR cells. The predictive value of the expression coordination for the expression regulation was validated with data from 8505C and BCPAP cell lines before and after lentiviral transfection with *DDX19B.*

## 1. Introduction

Thyroid cancer (**TC**) has a lower incidence and mortality rate compared to other malignancies. Still, in 2020 in the USA, 52,440 new cases (12,270 men and 40,170 women) are expected to be added. Although TC affects over three times more women than men, the number of deaths (2180) is practically equally distributed between the two sexes (1040 men and 1140 women) [1]. There are four major types of thyroid cancers: papillary (hereafter denoted as **PTC**, 70–80% of total cases), follicular (**FTC**, 10–15%), medullary (**MTC**, ~2%) and anaplastic (**APC**, ~2%). PTC, FTC and MTC are composed of well-differentiated cells and are treatable, while APC is undifferentiated and has a poor prognosis [2]. 

Considerable effort has been invested in recent decades to identify DNA mutations and the oncogenes (which turn on) and tumor suppressor factors (which turn off) that are responsible for triggering TC. The 25.0 release (22 July 2020) of the Genomic Data Commons Data Portal [3] includes 11,128 confirmed mutations detected on 13,564 genes sequenced from 1440 (553 male and 887 female) TC cases. The most frequently mutated gene reported in the portal is *BRAF* (B-Raf proto-oncogene, serine/threonine kinase), with up to 10 mutations identified in 20.56% of cases. Further down in terms of mutation frequency are: *NRAS* (neuroblastoma RAS viral (v-ras) oncogene homolog) with two mutations in 2.71% of cases, *TTN* (titin), with a total of 40 mutations in 2.29% of cases, and *TG* (thyroglobulin), with a total of 26 mutations in 1.67% of cases [3]. For most genes, the portal [3] shows the specific types and locations of the mutations and the cancer form(s) where these mutations were found. However, there is no bi-univocal correspondence between cancer forms and mutated genes: each cancer was associated with numerous mutated genes and mutations of the same gene were identified in several forms of cancer. How many mutated genes does one need in order to decide upon the right form of cancer? Are there exclusive combinations of mutations for a particular form of cancer and only for that form? If present, the number of the affected genes should be large enough to avoid any overlap with other form of cancer. Although the incidence of each particular mutation in the explored cohort of patients is known, it is impossible to determine the predictive values of combinations of mutated genes because for more than three genes the number of possibilities (≥2.3 × 10^11^) exceeds the human population of the Earth. Even though one can determine via conditioned probabilities (actual conditioned frequencies) the chance of finding the same combination of mutations in other persons, the diagnostic value is very poor. Moreover, one should not forget that the mutations were identified with respect to a reference human genome obtained by averaging the DNA sequence results from a large number of healthy individuals regardless of race, sex, age, environmental conditions, etc. However, even among genomes of healthy individuals there are 0.1% (i.e., ~3mln nucleotides) differences (0.6% when considering indels) [4]. 

There are several commercially available gene assays used for the preoperative diagnostic and classification of TCs (e.g., [5,6]). Recently, Foundation Medicine [7] compiled a list of 310 genes with full coding exonic regions for the detection of substitutions, insertion–deletions and copy-number alterations. An additional list of the same Foundation contains 36 genes with intronic regions useful for the detection of gene rearrangements (one gene with a promoter region and one non-coding RNA gene) [7]. For all these assays, the question is how many and what genes should be mutated/regulated to assign an accurate diagnostic? Most importantly, how did the researchers determine the predictive values of each combination of genes? In [7], there are 346 combinations of one gene, 59,685 of two, 6,843,880 combinations of three, 587 million of four and over 40 billion of five and more. Therefore, for practical reasons, only the most relevant three biomarkers, at most, are currently used, which considerably limits the diagnostic accuracy.

While the diagnostic value of mutations and/or regulations is doubtful, what about their use for therapeutic purposes? Is restoring the normal sequence/expression level of one biomarker enough to cure the cancer? Considering that the “trusted” biomarkers were selected from the genes with the most frequently altered sequence and/or expression level in large populations, this means that they are less protected by the cellular homeostatic mechanisms. The cells are supposed to invest energy to protect the sequence and expression level of genes, critical for their survival, proliferation, and integration in multicellular structures. The low level of protection indicates that biomarkers are minor players, and therefore the restoration of their structure/expression level may be of little consequence to the cancer cells.

While we do not see a genomic solution for the cancer diagnostic at present, we believe that our Gene Master Regulator (**GMR**) approach [8,9] is a reasonable alternative to the actual biomarker-oriented gene therapy. The GMR of a particular cell phenotype is the gene whose highly protected sequence/expression by the cellular homeostatic mechanisms regulates major functional pathways through expression coordination with many of their genes. In our cancer genomic studies [8,9,10], we found that the GMR of the cancer nodule is very low in the gene hierarchy of the surrounding cancer-free tissue of the tumor. For this reason, manipulation of the GMR’s expression is expected to selectively destroy cancer cells without affecting the normal ones much. 

In this report, we analyze previously published transcriptomic data [8] to quantify the cancer-related remodeling of major functional pathways in the PTC nodule with respect to the normal tissue of the resection margins (**NOR**) of a surgically removed thyroid tumor. The Gene Commanding Height (**GCH**) hierarchy and the GMRs are determined in both PTC and NOR, and the potential regulations of the apoptosis genes in response to the cancer GMR expression manipulation are predicted. The GCH scores of the top genes are compared to those of the most mutated genes in TC as well as those of the usually considered cancer biomarkers. Transcriptomic profiles of two standard TC cell lines before and after stable transfection with a gene were used to determine the predictive value of the expression coordination with that gene in untreated cells for the regulation in treated ones. The analysis presented here was derived from the Genomic Fabric Paradigm (GFP) that assigns three independent measures to each gene and considers the transcriptome as a multi-dimensional mathematical object [11].

## 2. Materials and Methods 

### 2.1. Gene Expression Data

We used gene expression data from one case of papillary thyroid carcinoma, pathological stage pT3NOMx, deposited in the Gene Expression Omnibus (GEO) of the National Center for Biotechnology Information (NCBI) [12] as GSE97001. In that study, the quarters of the most homogeneous 20-mm^3^ part of the frozen unilateral, single, 32.0-mm PTC nodule and four small pieces from the NOR of the same gland from the same patient were profiled separately. Thus, we got data from four biological replicas of each region. Since each human is subjected to a unique set of transcriptome-regulating factors (race, sex, age, medical history, environmental conditions, exposure to stress and toxins, etc.), the normal tissue surrounding the cancer nodule is a far better reference than tissues from other healthy persons. Expression values were normalized iteratively to the median of all quantifiable genes in all samples and transcript abundances were presented as multiples of the expression level of the median gene in each region. 

Transcriptomic data from the surgically removed tumor were compared to the gene expression profiles of two standard human thyroid cancer cell lines: BCPAP (papillary) and 8505C (anaplastic) deposited as GSE97002. We determined the predictive value of the coordination analysis in untreated cells for the expression regulation in treated ones by comparing the transcriptomic profiles of these cell lines before and after stable transfection with *DDX19B, NEMP1, PANK2* and *UBALD1*. The results of transfection with DEAD (Asp-Glu-Ala-Asp) box polypeptide 19B (*DDX17B*) were collected from GSE97028, those for nuclear envelope integral membrane protein 1 (*NEMP1*) from GSE97031, for pantothenate kinase 2 (*PANK2*) from GSE97030 and for UBA-like domain containing 1 (*UBALD1*) from GSE97427. Although alterations of *DDX19B* [13], *NEMP1* [14] and *PANK2* [15] were linked to some forms of cancer by other authors, these genes were selected only because their different GCH scores in the two cell lines made them suitable to validate the GMR approach [8,9].

### 2.2. Single-Gene Transcriptomic Quantifiers

#### 2.2.1. Biological Replicas, Profiling Redundancy and Average Expression Level

The four biological replicas experimental design provided for every single gene in each region three independent measures: (i) average expression level, (ii) expression variation and (iii) expression coordination with each other gene [16]. We used these three measures and combinations of them to establish the gene hierarchies and characterize the contribution of each gene to the cancer-related reorganization of the thyroid transcriptome. 

The Agilent two-color expression microarrays used in the analyzed experiment redundantly probed the genes with various number of spots from 1 to 20 (as for *MIEF1* = mitochondrial elongation factor 1) and *SRRT* = serrate, RNA effector molecule). Therefore, for each gene “*i*”, we computed the average expression level over the group of *R_i_* spots redundantly probing the same transcript of the average expression levels measured by spot “*k*” across the biological replicas.
(1)$$\begin{array}{l} \mu_{i}^{\left(NOR/PTC\right)}=\frac{1}{R_{i}}\sum_{k=1}^{R_{i}}\mu_{i,k}^{\left(NOR/PTC\right)}=\frac{1}{R_{i}}\sum_{k=1}^{R_{i}}\left(\frac{1}{4}\sum_{j=1}^{4}a_{i,k,j}^{\left(NOR/PTC\right)}\right),\;where:\\ a_{i,k,j}^{\left(NOR/PTC\right)}=\mathrm{expression}\;\mathrm{level}\;\mathrm{of}\;\mathrm{gene}\;“i”\;\mathrm{probed}\;\mathrm{by}\;\mathrm{spot}\;“k”\;\mathrm{on}\;\mathrm{biological}\;\mathrm{replica}\;“j” \end{array}$$

#### 2.2.2. Expression Variation

Because of the probing redundancy, instead of the coefficient of variation (CV), we used the *Relative Expression Variability* (*REV*). *REV* is the Bonferroni-like corrected mid-interval of the chi-square estimate of the pooled CV for all quantifiable transcripts of the same gene [17]
(2)$$\begin{array}{c} REV_{i}^{\left(NOR/PTC\right)}=\underset{\mathrm{correction}\;\mathrm{coefficient}}{\underset{︸}{\frac{1}{2}\left(\sqrt{\frac{r_{i}}{\chi^{2}\left(r_{i};0.975\right)}}+\sqrt{\frac{r_{i}}{\chi^{2}\left(r_{i};0.025\right)}}\right)}}\sqrt{\underset{\mathrm{pooled}\;\mathrm{CV}}{\underset{︸}{\frac{1}{R_{i}}\sum_{k=1}^{R_{i}}{\left(\frac{s_{ik}^{\left(NOR/PTC\right)}}{\mu_{ik}^{\left(NOR/PTC\right)}}\right)}^{2}}}}\times 100\%\\ \mu_{ik}=\mathrm{average}\;\mathrm{expression}\;\mathrm{level}\;\mathrm{of}\;\mathrm{gene}\;i\;\mathrm{probed}\;\mathrm{by}\;\mathrm{spot}\;k\;(=1,\;\ldots ,\;R_{i})\;\mathrm{in}\;\mathrm{the}\;4\;\mathrm{biological}\;\mathrm{replicas}\\ s_{ik}=\mathrm{standard}\;\mathrm{deviation}\;\mathrm{of}\;\mathrm{the}\;\mathrm{expression}\;\mathrm{level}\;\mathrm{of}\;\mathrm{gene}\;i\;\mathrm{probed}\;\mathrm{by}\;\mathrm{spot}\;k\\ r_{i}=4R_{i}-1=\mathrm{number}\;\mathrm{of}\;\mathrm{degrees}\;\mathrm{of}\;\mathrm{freedom}\\ R_{i}=\mathrm{number}\;\mathrm{of}\;\mathrm{microarray}\;\mathrm{spots}\;\mathrm{probing}\;\mathrm{redundantly}\;\mathrm{gene}\;i \end{array}$$

A lower *REV* indicates stronger control by the cellular homeostatic mechanisms to limit the expression fluctuations, expected for genes critical for survival, proliferation and phenotypic expression. Therefore, we also use the *Relative Expression Control* (*REC*)
(3)$$\begin{array}{l} REC_{i}^{\left(NOR,PTC\right)}\equiv \frac{{\langle REV\rangle }^{\left(NOR/PTC\right)}}{REV_{i}^{\left(NOR/PTC\right)}}-1\\ \langle \rangle =\mathrm{median}\;\mathrm{for}\;\mathrm{all}\;\mathrm{genes}\;\mathrm{profiled}\;\mathrm{in}\;\mathrm{that}\;\mathrm{phenotype} \end{array}$$

As defined, positive *REC*s point to genes that are more controlled than the median while negative *REC*s identify less controlled genes in that phenotype. It is natural to assume that the cell invests more energy to control the expressions of more important genes for its survival, phenotypic expression and integration into a multi-cellular structure. As such, *REC* is a major factor to consider in establishing the gene hierarchy.

#### 2.2.3. Expression Coordination

The expression coordination of two genes in the same region was quantified by their pair-wise momentum-product Pearson correlation coefficient between the two sets of expression levels across biological replicas, “$\rho_{ij}^{\left(NOR/PTC\right)}$”. The statistical significance was evaluated with the two-tail *t*-test for the degrees of freedom df = 4(biological replicas)*R (number of spots probing redundantly each of the correlated transcripts) − 2. Two genes were considered as synergistically expressed (positive or in-phase coordination) if their expression levels fluctuated in phase across biological replicas. They are considered as antagonistically expressed (negative or anti-phase coordination) when their expression levels manifest opposite tendencies and are independently expressed (neutral coordination) when the expression fluctuations of one gene are not related to the fluctuations of the other [17]. Although not (yet) validated through molecular biology studies, the expression coordination was speculated to reflect the “transcriptomic stoichiometry” of the encoded proteins that are produced in certain proportions to optimize the cellular functional pathways [18]. 

We also computed the coordination power $CP_{i,\Gamma }^{\left(NOR/PTC\right)}$ [19] and the Overall Coordination $OC_{i,\Gamma }^{\left(NOR/PTC\right)}$ of a gene “*i*” with respect to the functional pathway “*Γ*” in each of the two profiled regions (NOR and PTC)
(4)$$CP_{i,\Gamma }^{\left(NOR/PTC\right)}\equiv {\bar{\rho_{ij}^{\left(NOR/PTC\right)}}|}_{\forall j\in \Gamma .j\neq i}\times 100\%,\;OC_{i,\Gamma }^{\left(NOR/PTC\right)}\equiv \exp\left(\frac{4}{N}\sum_{j\in \Gamma ,j\neq i}\rho_{ij}^{2}-1\right)$$

Both $CP_{i,\Gamma }^{\left(NOR/PTC\right)}$ and $OC_{i,\Gamma }^{\left(NOR/PTC\right)}$ are measures of the gene “*i*” influence on “*Γ*”.

### 2.3. Gene Commanding Height (GCH) and Gene Master Regulator (GMR) 

In previous papers [8,9,10], we introduced the Gene Commanding Height (GCH), a combination of the expression control and expression coordination with all (ALL) other genes, to establish the gene hierarchy in each phenotype
(5)$$GCH_{i}^{\left(NOR,PTC\right)}\equiv \left(REC_{i}^{\left(NOR,PTC\right)}+1\right)OC_{i,ALL}^{\left(NOR/PTC\right)}\equiv \frac{{\langle REV\rangle }^{\left(NOR/PTC\right)}}{REV_{i}^{\left(NOR/PTC\right)}}\exp\left(\frac{4}{N}\sum_{j\in ALL,j\neq i}\rho_{ij}^{2}-1\right)$$

The top gene (highest GCH) in each phenotype was termed Gene Master Regulator (GMR) of that phenotype. The very strict control of the GMR expression suggests that this gene is utterly important for cell survival, while the very high overall coordination indicates how much its expression regulates the expression of many other genes. 

### 2.4. Expression Regulation

A gene was considered as significantly regulated in the PTC with respect to the NOR if the absolute expression ratio exceeds the cut-off (CUT) value computed individually for each gene by considering the expression variabilities of that gene in both compared conditions [9].
(6)$$\begin{array}{l} \left|x_{i}^{\left(NOR\to PTC\right)}\right|>CUT_{i}=1+\frac{1}{100}\sqrt{2\left({\left(REV_{i}^{\left(NOR\right)}\right)}^{2}+{\left(REV_{i}^{\left(PTC\right)}\right)}^{2}\right)},\;where:\\ x_{i}\equiv \{\begin{array}{cc} \frac{\mu_{i}^{\left(PTC\right)}}{\mu_{i}^{\left(NOR\right)}} & ,\;\mathrm{if}\;\mu_{i}^{\left(PTC\right)}>\mu_{i}^{\left(NOR\right)}\\ -\frac{\mu_{i}^{\left(NOR\right)}}{\mu_{i}^{\left(PTC\right)}} & ,\;\mathrm{if}\;\mu_{i}^{\left(PTC\right)}<\mu_{i}^{\left(NOR\right)} \end{array},\;\mu_{i}^{\left(PTC/NOR\right)}=\frac{1}{R_{i}}\sum_{k=1}^{R_{i}}\mu_{ik}^{\left(PTC/NOR\right)} \end{array}$$

The “CUT” criterion for individual genes eliminates the false positives and the false negatives selected by considering uniform absolute fold-change cut-off (e.g., 1.5*x*). In addition to the percentage of up- and down-regulated genes (that considers all genes as equal contributors to the alteration of a pathway), or the expression ratios “*x*”, we prefer the Weighted Individual (gene) Regulation [20], “*WIR*”:(7)$$\begin{array}{l} WIR_{i}^{\left(NOR\to PTC\right)}\equiv \mu_{i}^{\left(NOR\right)}\frac{x_{i}}{\left|x_{i}\right|}\left(\left|x_{i}\right|-1\right)\left(1-p_{i}\right)\;where:\\ \mu_{i}^{\left(NOR\right)}=\mathrm{average}\;\mathrm{expression}\;\mathrm{in}\;\mathrm{the}\;\mathrm{normal}\;\mathrm{tissue},\\ p_{i}=p\text{-}\mathrm{value}\;\mathrm{of}\;\mathrm{the}\;\mathrm{regulation} \end{array}$$

Note that in Equation (7), WIR takes into account the normal expression of that gene (i.e., in NOR), its expression ratio (PTC vs. NOR) and the confidence interval (1-p) of the regulation. 

### 2.5. Quantifiers of the Functional Genomic Fabrics

The Kyoto Encyclopedia of Genes and Genomes [21,22] was used to select the genes involved in the thyroid hormone synthesis (THS), cell cycle (CC) and oxidative phosphorylation (OPH), as well as how experimental manipulation of the PTC GMR might regulate the programmed cell death (apoptosis, APO). Although almost all functional pathways were perturbed in cancer, THS, CC, OPH and APO were selected because of their importance for the thyroid function and cancer development. There are reports of altered THS in cancer progression and apoptosis (e.g., [23]) and the role of the thyroid hormone in regulating the cell-cycle [24] and the oxidative phosphorylation [25].

Median REC over a gene selection (e.g., apoptosis pathway) was used to compare the expression controls of that selection in different regions or two different gene selections in the same region. Alteration of the genomic fabrics was quantified by the average “*X*” of the absolute expression ratios and by Weighted Pathway Regulation (WPR), the average of the absolute *WIR*s over a particular “selection” of genes
(8)$$\begin{array}{l} X_{selection}^{\left(NOR\to PTC\right)}\equiv {\bar{\left|x_{i}^{\left(NOR\to PTC\right)}\right|}}_{\forall i\in selection}\\ WPR_{selection}^{\left(NOR\to PTC\right)}\equiv {\bar{\left|WIR_{i}^{\left(NOR\to PTC\right)}\right|}}_{\forall i\in selection} \end{array}$$

## 3. Results

### 3.1. Overall Results

A total of 14,903 well-quantified unigenes in all PTC and NOR samples, and in BCPAP and 8505C cells before and after transfection with one of the four targeted genes, were considered in the sequent analyses. The groups redundantly probing the same transcript were replaced by their averages in each biological replica. Eukaryotic translation elongation factor 1 α 1 (*EEF1A1*) had the largest expression (82.31 median gene expression units) in NOR (not significantly regulated in PTC). Niemann-Pick disease, type C2 (*NPC2*) had the largest expression in PTC (86.97 median gene expression units), up-regulated by 7.24x with respect to NOR. Notch 1 (*NOTCH1*) with 82.35 had the largest expression in the BCPAP cells and myelin protein zero-like 3 (*MPZL3*) with 107.60 tops the gene expression level in the 8505C cells.

Out of the quantified unigenes, 1225 (8.22%) were down-regulated and 1852 (12.42%) were up-regulated in PTC with respect to NOR. The average absolute PTC/NOR expression ratio for all genes was *X* = 1.768 (median |x| = 1.309) and the *WPR* was 1.071 (median WIR = 0.046). Chitinase 3-like 1 (*CHI3L1*) was the most up-regulated (x = 219.38) and trefoil factor 3 (intestinal) (*TFF3*) the most down-regulated (x-99.86) gene in PTC. Because expression coordination and average expression level are independent measures, the high regulation of these genes in PTC with respect to NOR has no relevance for their networking in either of the two profiled regions. 

### 3.2. Three Independent Measures for Each Gene

Figure 1 illustrates the independence of the three measures for the first 50 alphabetically ordered genes involved in the KEGG-derived [22] human apoptosis pathway (hsa04210). We chose *IL6* (interleukin 6) to illustrate the expression coordination of apoptotic genes owing to the significant role of the encoded protein (IL6) in the PTC development [26]. However, coordination with any other gene supports the same conclusion. In addition to the clear independence of the three measures, transcriptomic differences between the two histo-pathologically distinct profiled regions from the thyroid are evident. 

In this gene selection, FBJ murine osteosarcoma viral oncogene homolog (*FOS*) has the highest average expression level (45.37) in NOR (significantly down-regulated by −2.06x in PTC). Cathepsin H (*CTSH*) had the largest expression (35.23), up-regulated by 6.78x with respect to NOR. *FOS,* cathepsin K (*CTSK*) and inhibitor of kappa light polypeptide gene enhancer in B-cells kinase γ (*IKBKG*) were among the significantly down-regulated genes. In contrast, *BID* (H3 interacting domain death agonist), *CTSH* and *DIABLO* (diablo, IAP-binding mitochondrial protein), were among the up-regulated genes of the selection. 

CASP8 and FADD-like apoptosis regulator (*CLFAR*) was the most variably expressed gene in the normal tissue and DNA fragmentation factor (*DFFB*), 40kDa, β polypeptide (caspase-activated DNase) the most variably expressed in PTC. Note that most of the selected genes have larger expression variability in the normal tissue than in the cancer nodule. This result confirms our previous reports (see Discussions) about diseases triggering increased control exerted by the cellular homeostatic mechanisms on the transcripts abundances as a way to protect against extensive damages. 

Observe also that 20 (40%) of the illustrated apoptotic genes are synergistically expressed with *IL6* in the normal tissue and only two (4%) in the PTC nodule, suggesting the decoupling of the programmed cell death from the inflammatory response in cancer.

### 3.3. Expression Regulation

Figure 2 illustrates the contributions of the first 50 alphabetically ordered quantified oncogenes to the overall regulation in PTC measured by the percentages of the up- and down-regulated genes, expression ratios and weighted individual (gene) regulations. The percentages are restricted to only the significantly regulated genes (considered as equal −1/+1 contributors). By contrast, both *X* and WIR take into account all (regulated and not regulated) genes and the contributions of these genes are no longer uniform. More informative than the expression ratio, WIR weights the contribution of each gene by its normal expression level (i.e., in NOR), fold-change in cancer and statistical significance of the regulation. 

### 3.4. Regulation of the Thyroid Hormone Synthesis

Figure 3 presents the regulations of the genes involved in the (KEGG-determined) thyroid hormone synthesis (hsa04918). In this pathway, 10.0 (20%) of the 50 quantified genes were up-regulated and six (12%) were down-regulated.

### 3.5. Regulation of the Cell-Cycle Pathway

Figure 4 presents the regulation of the genes involved in the (KEGG-determined) cell cycle pathway (hsa04110), where, out of the 93 genes quantified, three (3.23%) were down-regulated and 14 (15.05%) were up-regulated. Except *PTTG2*, all other regulated genes are located in the DNA replication (S-phase) and the two temporal gaps, G1 and G2, separating the S phase from mitosis (M-phase), indicating faster replication but stationary differentiation. 

### 3.6. Remodeling of the Oxidative Phosphorylation Pathway

Figure 5 presents the remodeling of the coordination networks interlinking the five complexes ([C1], [C2], [C3], [C4], [C5]) of the oxidative phosphorylation in the PTC nodule with respect to NOR tissue. The genes were selected from the KEGG hsa 00190. Note the substantial increase in the synergistically expressed gene pairs in PTC (273) with respect of the NOR (155) and that there is no antagonistically expressed gene pair in PTC, while in NOR there are 105. When the coordination inside each complex is added, there are 781 synergistic and 0 antagonistic pairs in PTC versus 458 synergistic and 242 antagonistic pairs in NOR. In addition to the eight up-regulated and three down-regulated genes within the selection of the 92 oxidative-phosphorylation genes, these results indicate a significant increase in the coordination of the complexes involved in the OP activity. 

### 3.7. Gene Hierarchy

Figure 6 presents the GCH scores of the 12 most frequently mutated genes in TC (reported in [1]) and the top 12 genes in NOR and PTC. Mutated genes: B double prime 1, subunit of RNA polymerase III transcription initiation factor IIIB (*BDP1*), B-Raf proto-oncogene, serine/threonine kinase (*BRAF*), *DST* (dystonin), eukaryotic translation initiation factor 1A, X-linked (*EIF1AX*), Harvey rat sarcoma viral oncogene homolog (*HRAS*), lysine (K)-specific methyltransferase 2A (*KMT2A*), microtubule-actin crosslinking factor 1 (*MACR1*), metastasis associated lung adenocarcinoma transcript 1 (non-protein coding) (*MALAT1*), neuroblastoma RAS viral (v-ras) oncogene homolog (*NRAS*), thyroglobulin (*TG*), ubiquitin-specific peptidase 9, X-linked (*USP9X*), zinc finger homeobox 3 (*ZFHX3*). Note that none of the most frequently mutated genes are among the top 12 genes in either region. Even *BRAF*, mutated in 20.56% of the 1440 cases, has no competitive GCH to be a good candidate for the PTC gene therapy (GCH of *BRAF* in PTC is 11.79). However, *SPINT2,* the PTC’s GMR ($GCH_{SPINT2}^{\left(PTC\right)}$ = 54.97), appears to be the most legitimate target for this case. While significant alteration of the expression of *SPINT2* would have lethal impact on the cancer cells, due to the very low GCH in NOR ($GCH_{SPINT2}^{\left(NOR\right)}$ = 1.93), it might have very little consequences on the normal cells. Importantly, the GCH scores of the top genes in PTC are substantially lower in NOR and vice-versa.

For comparison, we added the GCH scores of the top 23 genes in each of the standard TC cell lines BCPAP (papillary) and 8505C (anaplastic). Remarkably, 14 genes in the BCPAP cells and three genes in the 8505C cells have GCH scores higher than *SPINT2* in PTC. As an additional reference, Appendix A shows the GCH scores of most of the genes from FoundationOne^®^CDx (Foundation Medicine, Cambridge, MA, U.S.A.) used by Foundation Medicine [7] for genomic testing of solid tumors, including “Non-Small Cell Lung (NSCLC), Colorectal, Breast, Ovarian, and Melanoma. The list contains genes with full coding exonic regions for the detection of substitutions, insertion-deletions (indels), and copy-number alterations (CNAs). It also includes genes with select intronic regions for the detection of gene rearrangements, one gene with a promoter region (telomerase reverse transcriptase (*TERT*)) and one non-coding RNA gene (*TERC*). These genes might be useful for diagnostic purposes. However, with their GCH score far below the GMR’s and with not enough difference between PTC and NOR, they should have little therapeutic value for this particular case. Substantially lower than the PTC GMR were the biomarkers, oncogenes, apoptosis genes and the ncRNAs determined in the same specimens and presented in Figure 2 from [8].

### 3.8. The Gene Master Regulator at Play

Our study identified *SPINT2,* a not regulated gene in the investigated PTA, as the GMR of this patient’s malignancy. What are the mechanisms by which experimental alteration of *SPINT2* expression might selectively kill the cancer cells but not the normal ones? *SPINT2* is highly coordinated with numerous genes from almost all major functional pathways. However, we considered that apoptosis might be the best candidate to evaluate (from a bioinformatics point of view) the effects of *SPINT2* manipulation. Therefore, we analyzed the expression coordination of *SPINT2* with the 112 apoptosis genes quantified in the two regions. Table 1 presents the apoptosis genes that are significantly up/down regulated in PTC or/and significantly synergistically/antagonistically/independently expressed with *SPINT2* in NOR or/and PTC. 

In PTC (21 significantly up-regulated andseven down-regulated apoptosis genes), we found *SPINT2* to be significantly synergistically expressed with 34 apoptosis genes, but with no significant antagonistically or independently expressed partners. This is a substantial increase from the six synergistically, one antagonistically and four independently expressed apoptosis partners of *SPINT2* in NOR. Interestingly, none of the significant correlations in NOR (with: *ACTG1, ATF4, CASP9, CTSL, CYCS, NFKB1*) were maintained in PTC.

What effect the overexpression of an otherwise stably expressed but not regulated gene in PTC (*SPINT2*) may have on cancer cells? Most probably, owing to the substantial expression synergistic coordination with apoptosis genes, the experimental overexpression of *SPINT2* would up-regulate many of these genes, forcing the commanded (PTC) cells to enter programmed death.

There was no way to validate this hypothesis on the patient from whom we had profiled the thyroid tumor. However, we tested the general hypothesis that expression coordination with one gene predicts expression regulation when the expression of that gene is experimentally manipulated. For this purpose, we analyzed the transcriptomes of the TC cell lines BCPAP and 8505C cells before and after stable lentiviral transfection with either *DDX19B, NEMP1, PANK2* or *UBALD1* [8]. Figure 7a,b plots the correlation coefficient with *DDX19B* in the untreated cells against the fold-changes (negative for down-regulation) of the genes in the transfected cells. They clearly show that expression coordination predicts (>86%) the expression regulation with reasonable accuracy. Similar validation (83–91%) was obtained for the same cell lines transfected with either *NEMP1, PANK2* or *UBALD1.* Based on this validation, Figure 7c illustrates the predicted regulations of apoptosis genes if the expression level of *SPINT2* in PTC is significantly increased. 

In Figure 7c, we used the uniform contribution of the significantly altered genes to the percentages of (up-/down-) regulated genes. Note that from 21 up-regulated and six down-regulated genes in untreated PTC, overexpression of *SPINT2* may result in 48 up-regulated and six down-regulated genes. The expression of six genes, *BBC3, DAB2IP, DIABLO, PIK3R2, PMAIP1, TNFRSF1A*, which are already up-regulated in PTC may be further increased by treatment, while the down-regulation of *GZMB* in untreated PTC may be recovered by overexpressing *SINT2*.

## 4. Discussion

Although with no molecular biology validation, the bioinformatics analysis of the gene expression profiles in the cancer nodule and surrounding normal tissue of a surgically removed papillary tumor produced some very interesting results, out of which the most important are: Each cell phenotype from the tumor is governed by a different gene hierarchy and a distinct organization of its transcriptome;As selected from the most altered genes in a large population of cancer patients, the biomarkers have low GCH and therefore little therapeutic value;The GMR of the cancer nodule is the most legitimate target of the gene therapy because it is the most influential gene for cancer cells while having very little role in the surrounding normal cells;*SPINT2* was identified as the GMR of the PTC nodule of the profiled tumor and a gene with very low GCH score in NOR;The up-regulation of the synergistically expressed apoptosis genes in untreated PTC following the experimental *SPINT2* overexpression was identified as a potential mechanism of selectively killing the cancer cells.

The analysis presented in this report is consistent with the genomic fabric paradigm [11] that considers the transcriptome as a multi-dimensional object subjected to a dynamic set of expression correlations among the genes. The traditional transcriptomic analysis is limited to the expression level of individual genes and comparisons of the expression levels of distinct genes in the same condition or of the same gene in different conditions. Our procedure considerably enlarges the transcriptomic information by considering for each gene not one, but three independent features and all possible combinations of these features to compare the genes and groups of genes in the same condition or across various conditions. 

Although high levels of *EEF1A1* were reported in renal cell carcinoma [27], we found this gene to have the highest expression in NOR and one of the highest levels in PTC (68.47), albeit not significantly down-regulated. A high expression of *NPC2* and its significant elevation in PTC were also detected in meta-analyses of public PTC transcriptomes [28]. Overexpression of *CHI3L1*, the most up-regulated gene in the analyzed PTC, was reported as associated with metastatic PTC [29] and its recurrence [30]. Significantly decreased expression of *TFF3*, the most down-regulated gene in our study, was also reported in several other studies (e.g., [31]).

In addition to illustrating the independence of the three features, Figure 1 provides also some interesting findings and confirmations of results reported by other authors. For instance, the high expression of *CTSH* in PTC (up-regulated by 6.78x with respect to NOR) was related to the tumor progression and migration of cancer cells [32]. 

The median REV has a statistically significant (*p*-value = 7.79 × 10^−5^) decrease from 39.75 in NOR to 38.69 in PTC. According to the Second Law of Thermodynamics, the significantly larger overall expression variability in NOR than in PTC indicates not only relaxed control by the homeostatic mechanisms (average REC_NOR_ = 0.084, average REC_PTC_ = 0.113), but also that NOR is closer to the thermodynamic (here physiological) equilibrium. Supporting this assertion is the reduction in the median REV observed by us in all other gene expression studies on animal models of human diseases (e.g., [33,34,35]) and in tissues of animals subjected to various stresses (e.g., [36,37,38] or to genetic manipulations (e.g., [39,40]). The high expression variability of *CFLAR* (a key anti-apoptosis regulator [41]) in NOR (REV = 102.93) may explain the adaptability of the apoptosis pathway to a large spectrum of environmental conditions. The *CFLAR* REV dramatic reduction in PTC (REV = 29.41) shows the need for tighter control of resisting apoptosis in cancer. Moreover, its reduction in expression level (in PTC by −1.64x) was associated with delayed apoptosis [41].

The observed down-regulation of *FOS* in PTC (Figure 2) confirms the findings of some groups [42,43] but contradicts its frequent (however not 100%) up-regulation reported by another group in 40 patients with thyroid cancer and 20 with benign thyroid diseases [44]. Let us analyze what measure of regulation is the most informative and use the examples of *FOS* and *KIT* (another gene down-regulated in thyroid cancer [45]). Although both genes account as units for the percentage of the down-regulated genes, they contribute −2.06x and −8.15x as expression ratios and −46.28 and −3.76 as WIRs. Since WIR is a more comprehensive measure, *FOS* regulation appears to be the most important factor in the alteration of this group of genes. Indeed, FOS protein is an important player in cell proliferation, differentiation, transformation and apoptotic cell death.

Among the significantly regulated genes from the KEGG-derived THS pathway (Figure 3), only *PAX8* (−1.94x) was previously related to the thyroid cancer, albeit to the follicular form. Moreover, we found that peroxisome proliferator-activated receptor γ (*PPARG*) whose fusion with *PAX8* is considered an important trigger of the FTC [46], was likewise significantly down-regulated (−4.99x). Interestingly, in Figure 3, the two glutathione peroxidases, *GPX1* (1.60x) and *GPX3* (−1.84x), were oppositely regulated. Since the down-regulation of *GPX1* was reported to augment the pro-inflammatory cytokine-induced redox signaling and endothelial cell activation [47], one may assume that up-regulation of *GPX1* will do the opposite, i.e., diminish the pro-inflammatory cytokine-induced redox signaling. As such, the PTC cells will become more resistant to the inflammatory response. 

According to the KEGG map hsa04110, some of the regulated cell-cycle genes (Figure 4) were associated with a wide diversity of cancers. Thus, up-regulation of *CDKN1A* was associated with cervical cancer [48] and down-regulation of *CDKN1C* with gastric cancer [49]. As stated in [3], up-regulation/mutation of *CDKN2A* was detected in numerous cancer forms: neoplasms (squamous cell, ductal, lobular, cystic, mucinous, serous, mesothelial, lipomathous, myomathous, thymic epithelial, complex, mixed), adenomas, adenocarcionmas, gliomas, nevi and melanomas, transitional cell papillomas and carcinomas, mature B-cell lymphomas, soft tissue tumors and sarcomas. *CDKN2B* was associated with malignant pleural mesothelioma, osteosarcoma and meningioma. However, we found no report associating these genes with thyroid cancer. Unfortunately, one of the most cancer-related genes, *TP53* [50], was not quantified in this experiment due to the corrupted probing spot in one of the microarrays, which can be seen as a major limitation of the study.

As illustrated in Figure 5 for interlinks between the five complexes of the oxidative phosphorylation, cancer remodels the gene networks, profoundly perturbing the mitochondrial function [51]. Among others, 10 synergistically expressed gene pairs in NOR are switched into antagonistically expressed pairs in PTC: *NDUFA10-SDHD, CYC1-COX1*, *COX10-ATP6V0B, COX10-ATP6V0C, COX5B-LHPP, COX6A1-ATV1B2, COX6A1-LHPP, COX6A2-ATP6V0B, COX6A2-ATP6V1A, COX7C-ATP6V1G2.* These switches, the cancellation of all significant antagonisms and the added synergisms increase the expression synchrony of the pathway genes [17] and remove the controlling bottlenecks. In a synergistic pair, the up-regulation of one gene triggers the up-regulation of the other. Although in this experiment we did not detect significantly altered expressions of *NDUFA10* and *SDHD*, their significant synergism in PTC may explain why they are both up-regulated in oral cancer [52].

Given the never-repeatable set of risk factors, each patient is unique and therefore their gene hierarchy is unique. Although the chance of finding the same GMR in two persons is about 1/20,000 and that of the first two genes is 1/400 million, from the first three genes up, the number of possibilities (7.9988 × 10^12^) exceeds by far the Earth human population. Therefore, the top three genes are enough to uniquely represent the cancer of each person at a given time. In our studied PTC, the top three genes were: *SPINT2*, *RPAP3* and *BZW,1* with none of them significantly regulated with respect to NOR.

*SPINT2*, the identified GMR of the profiled PTC (Figure 6), was previously reported by several groups to be involved in the development and progression of a wide diversity of forms of cancer [53]. Among others, *SPINT2* was associated with metastatic osteosarcoma [54], ovarian cancer [55], glioma/glioblastoma [56,57], prostate cancer [58] and non-small lung cancer [59], leukemia [60] and cervical carcinoma [61]. *RPAP3*, essential for assembling chaperone complexes [62], was linked to hypoxia-adapted cancer cells [63] and *BZW,1* was associated with ovarian [64], lung [65] and salivary gland [66] cancers. However, we found no mention in the literature about the role of these first three genes in any form of thyroid cancer.

In Figure 7 and Table 1, we tested whether expression synergism with apoptosis genes may be one of the mechanisms by which manipulation of *SPINT2* expression is lethal to the PTC cells but not to the NOR cells. First, we determined the significant coordination of *SPINT2* with apoptosis genes in both NOR and PTC and found a substantial increase in the expression synergism in PTC. Then, we tested the predictive value of the expression coordination by profiling two standard human TC cell lines before and after stable transfection with four genes selected, only to have substantially different GCHs in the two cell lines. Although *DDX19B* and *PANK2* (but not *NEMP1* and *UBALD1*) were synergistically expressed with *SPINT2* in PTC*,* there are no reports relating these genes with *SPINT2* in any form of cancer. As mentioned in [8], *NEMP1* and *PANK2* had higher GCHs and induced larger transcriptomic alterations in the BCPAP than in the 8505C cells. In contrast, *DDX19B* and *UBALD1* had higher GCHs and induced larger transcriptomic alterations in the 8505C than in the BCPAP cells. Figure 7a,b confirms our previous findings that expression correlation with one gene predicts what genes are regulated when the expression of that gene is manipulated. A similar conclusion was drawn in [67], where we had shown that most genes are synergistically/antagonistically expressed with *Gja* (encoding the gap junction protein Cx43) in the brain and hearts of wildtype mice are down-/up-regulated in the brain and hearts of Cx43KO mice. Therefore, as illustrated in Figure 7c, we expect that, due to the synergism, the overexpression of *SPINT2* will force the PTC cells into programmed death by up-regulating numerous apoptosis genes. 

## 5. Conclusions

Owing to the matchless set of conditioning factors, each human is unique and, despite all similarities, the transcriptomes of one person’s cell phenotypes can never be identical with those of another person. In a profiled metastatic clear cell renal cell carcinoma [34], we found that even the transcriptomes of two cancer nodules isolated from the same kidney and categorized with the same Fuhrman grade 3 were largely different from each other. Moreover, some of the gene expression conditioning factors (environment, exposure to stress and toxins, medical treatment, diet, ageing etc.) are not constant, forcing the transcriptomes of cancer cells to continuously adapt. By consequence, the gene hierarchy is not only unique for each person and in each of his/her cancer nodules, but it changes over time. As such, this study provides strong reasons in favor of a really personalized and time-sensitive cancer gene therapy based on the manipulation of the gene master regulators. 

## Figures and Tables

**Figure 1 genes-11-01030-f001:**
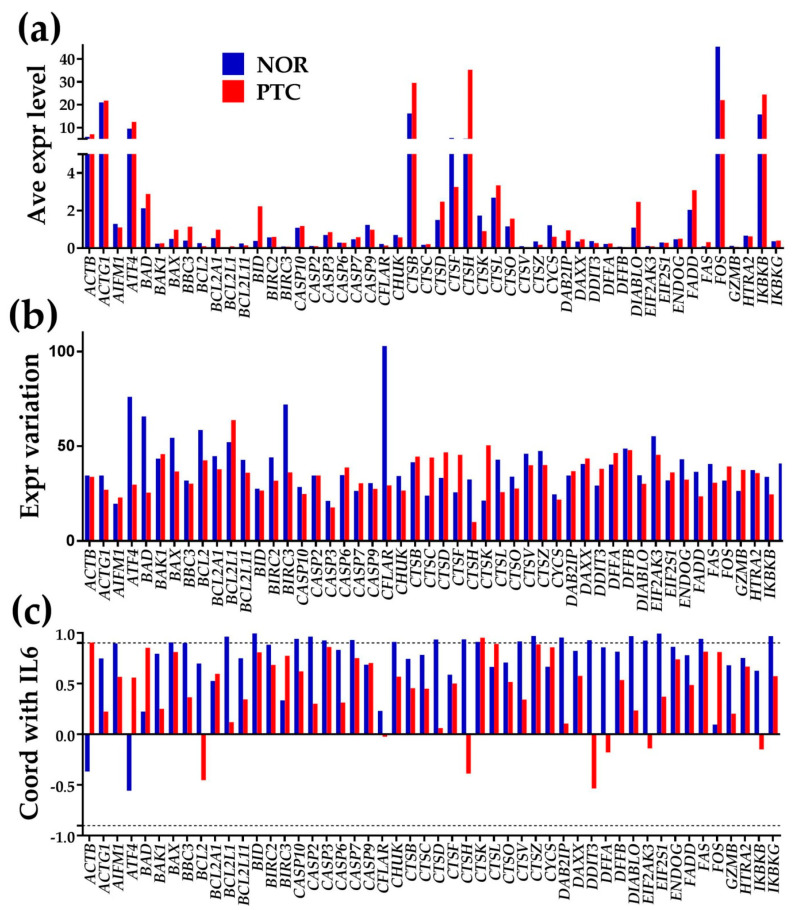
Three independent characteristics of every gene in each region. (**a**) average expression level, (**b**) expression variation and (**c**) expression coordination (here with IL6). The dashed black lines in panel (**c**) indicate the interval out of which the positive/negative coordination is considered as statistically significant.

**Figure 2 genes-11-01030-f002:**
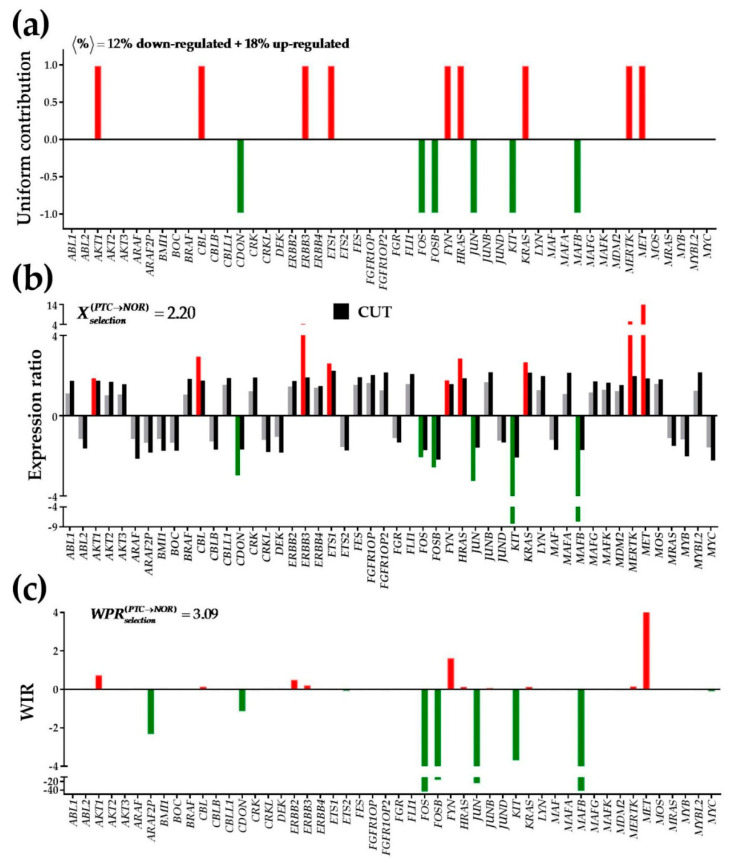
Three ways to consider the contribution of a gene to the pathway regulation. (**a**) uniform; (**b**) by expression ratio; (**c**) as Weighted Individual (gene) Regulation. Red/green/grey columns indicate up-/down-/not regulated genes. Black columns are the fold-change cut-offs (negative for down-regulation). Regulated genes: v-akt murine thymoma viral oncogene homolog 1 (*AKT1*), Cbl proto-oncogene, E3 ubiquitin protein ligase (*CBL*), cell adhesion associated, oncogene regulated (*CDON*), v-erb-b2 avian erythroblastic leukemia viral oncogene homolog 3 (*ERBB3*), v-ets avian erythroblastosis virus E26 oncogene homolog 1 (*ETS1*), homologs of FBJ murine osteosarcoma viral oncogene (*FOS/FOSB*), FYN proto-oncogene, Src family tyrosine kinase (*FYN*), Harvey rat sarcoma viral oncogene homolog (*HRAS*), jun proto-oncogene (*JUN*), Kirsten rat sarcoma viral oncogene homolog (*KRAS*), v-yes-1 Yamaguchi sarcoma viral related oncogene homolog (*LYN*), v-maf avian musculoaponeurotic fibrosarcoma oncogene homolog B (*MAFB*), c-mer proto-oncogene tyrosine kinase (*MERTK*) and met proto-oncogene (*MET*).

**Figure 3 genes-11-01030-f003:**
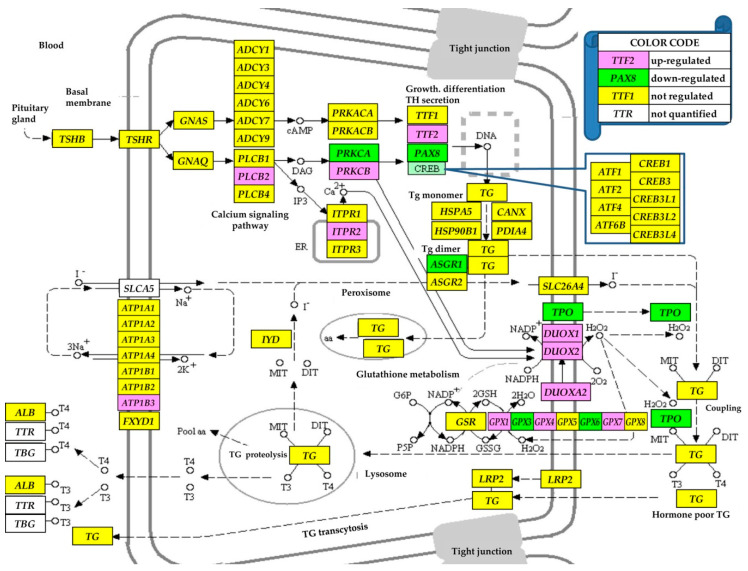
Regulation of thyroid hormone synthesis pathway (modified from hsa04918). Regulated genes: asialoglycoprotein receptor 1 (*ASGR1*), ATPase, Na+/K+ transporting, β 3 polypeptide (*ATP1B3*), dual oxidases (*DUOX1/2*), dual oxidase maturation factor 2 (*DUOXA2*), glutathione peroxidases (*GPX1/3/4/6/7*), inositol 1,4,5-trisphosphate receptor, type 2 (*ITPR2*), paired box 8 (*PAX8*), protein kinases C (*PRKCA/B*), thyroid peroxidase (*TPO*) and transcription termination factor, RNA polymerase II (*TTF2*).

**Figure 4 genes-11-01030-f004:**
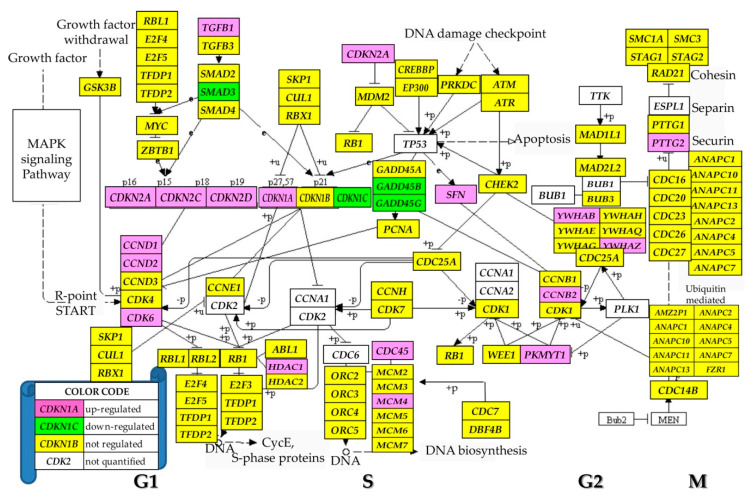
Regulation of the KEGG-determined cell cycle (hsa04110). Regulated genes: cyclins (*CCNB2/D1/D2*), cell division cycle 45 (*CDC45*), cyclin-dependent kinase inhibitors (*CDKN1A/1C/2A/2C/2D*), growth arrest and DNA-damage-inducibles (*GADD45B/D*), histone deacetylase 1 (*HDAC1*), minichromosome maintenance complex component 4 (*MCM4*), membrane associated tyrosine/threonine 1 (*PKMYT1*), pituitary tumor-transforming 2 (*PTTG2*), stratifin (*SFN*), SMAD family member 3 (*SMAD3*), transforming growth factor, β 1 (*TGFB1*) and tyrosine 3-monooxygenase/tryptophan 5-monooxygenase activation proteins (*YWHAB/Z*).

**Figure 5 genes-11-01030-f005:**
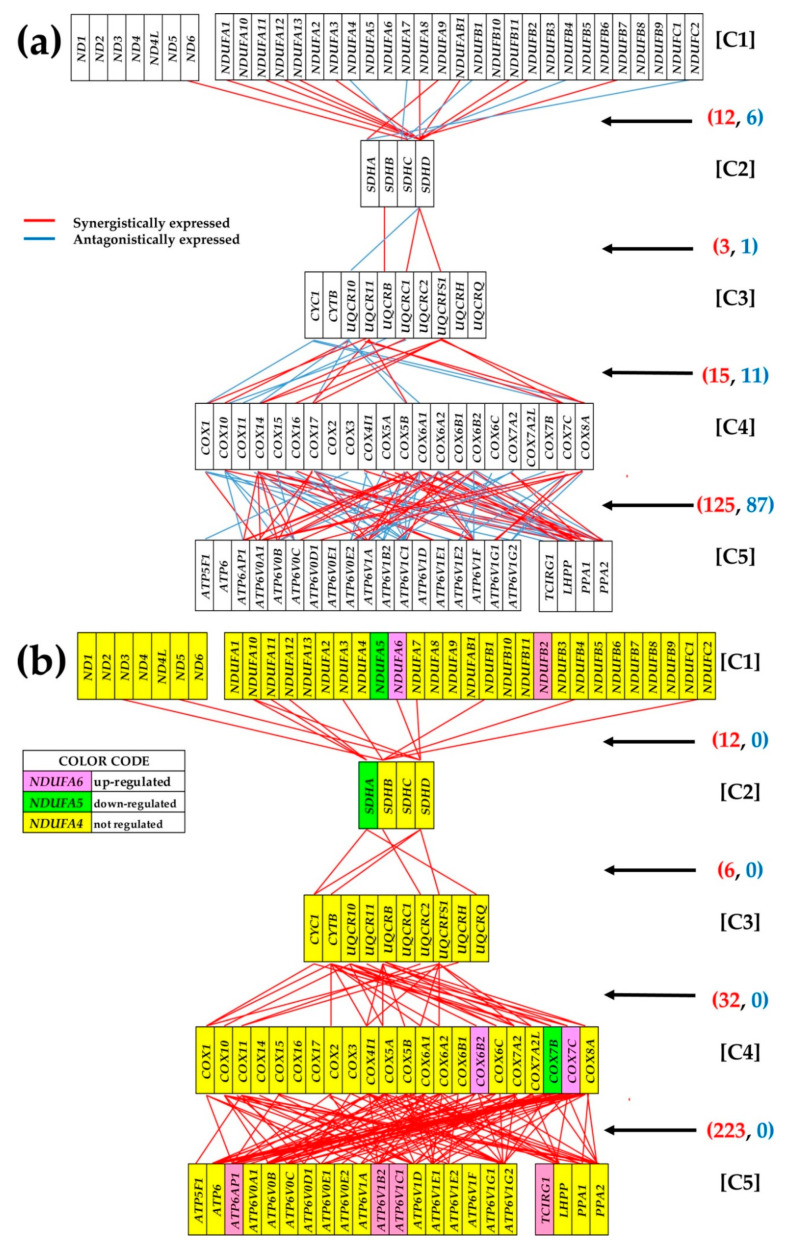
Remodeling of the coordination networks among the five complexes of the oxidative phosphorylation in the PTC nodule with respect to NOR tissue. The red/blue lines indicate that the connected genes are synergistically/antagonistically expressed in that region. Red/blue numbers in parentheses indicate the number of synergistically/antagonistically expressed gene pairs between the two complexes. Regulated genes: ATPase, H+ transporting, lysosomal proteins (*ATP6AP1, ATP6V1B2, ATP6V1C1*), cytochrome c oxidase subunits (*COX6B2, COX7B, COX7C*), NADH dehydrogenase (ubiquinone) 1 α/β subcomplexes (*NDUFA5, NDUFA6, NDUFB2*), succinate dehydrogenase complex, subunit A, flavoprotein (Fp) (*SDHA*) and T-cell, immune regulator 1, ATPase, H+ transporting, lysosomal V0 subunit A3 (*TCIRG1*).

**Figure 6 genes-11-01030-f006:**
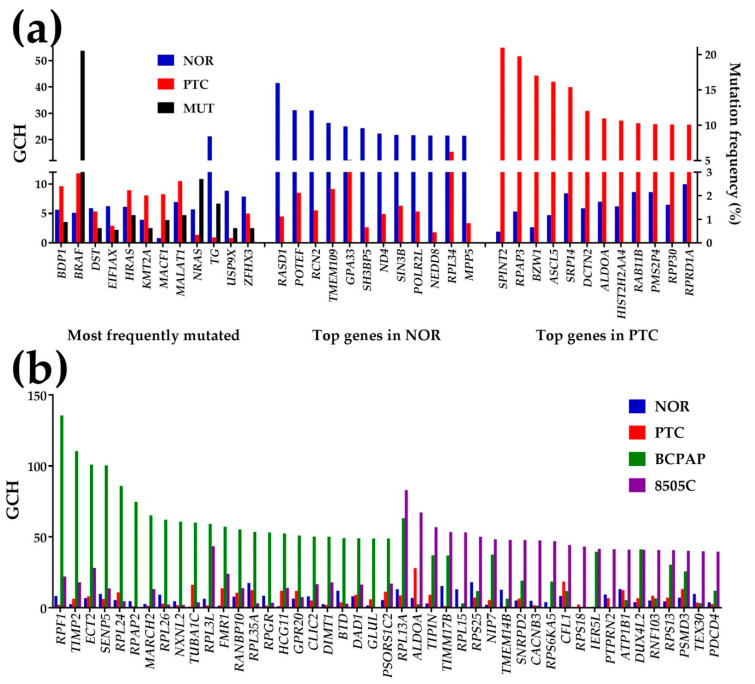
Gene Commanding Height (GCH). (**a**) GCH and mutation frequency of the 12 reported most frequently mutated genes and the top 12 genes in the normal tissue (NOR) and the papillary nodule (PTC). The mutation frequency is plotted on the right axis. (**b**) GCH of the top 23 genes in the papillary (BCPAP) and anaplastic (8505C) thyroid cancer cell lines and their scores in NOR and PTC. Top 3 genes in NOR: RAS, dexamethasone-induced 1 (*RASD1*), POTE ankyrin domain family, member F (*POTEF*), reticulocalbin 2, EF-hand calcium binding domain (*RCN2*). Top 3 genes in PTC: serine peptidase inhibitor, Kunitz type, 2 (*SPINT2*), RNA polymerase II associated protein 3 (*RPAP3*), basic leucine zipper and W2 domains 1 (*BZW1*). Top 3 genes in BCPAP cells: ribosome production factor 1 homolog (S. cerevisiae) (*RPF1*), TIMP metallopeptidase inhibitor 2 (*TIMP2*), epithelial cell transforming 2 (*ECT2*). Top 3 genes in 8505C cells: ribosomal protein L13a (*RPL13A*), aldolase A, fructose-bisphosphate (*ALDOA*), TIMELESS interacting protein (*TIPIN*).

**Figure 7 genes-11-01030-f007:**
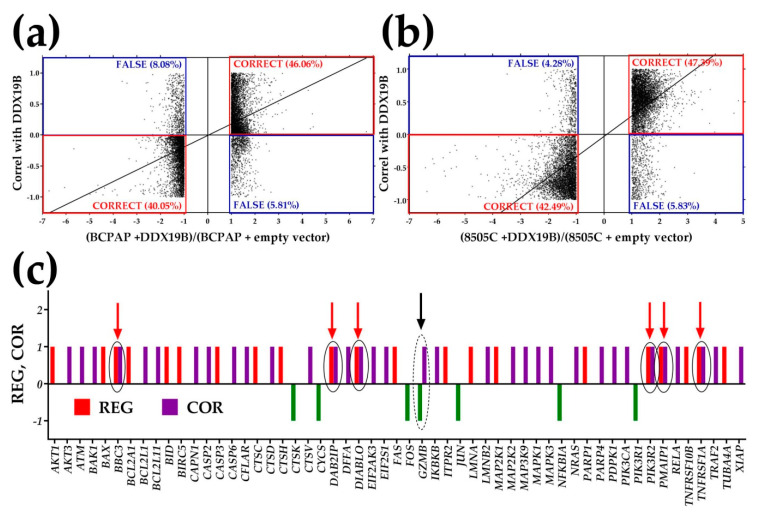
Prediction of the ripple effects of experimental gene regulation. (**a**) Expression coordination with *DDX19B* in untreated BCAP cells accurately predicts 86.11% (40.05 + 46.06) the type of the expression regulation in BCAP cells stably transfected with *DDX19B*; (**b**) Expression coordination with *DDX19B* in untreated 8505C cells accurately predicts 89.88% (42.49 + 47.39) of the type of the expression regulation in 8505C cells stably transfected with *DDX19B*; (**c**) Predicted regulation (1 for up-regulation and −1 for down-regulation) of apoptotic genes in PTC following experimental overexpression of *SPINT2.* REG = significant (1 = up-regulation, −1 = down-regulation). COR = significant expression synergism. Only the regulated genes in the untreated PTC and those expected to be regulated in treated PTC are represented. Red arrows indicate combined effect in treated tumor of regulation and expression synergism in untreated PTC. The black arrow indicates the down-regulation in untreated PTC expected to be compensated by the overexpression of *SPINT2.*

**Table 1 genes-11-01030-t001:** Apoptosis genes that are significantly up(U)/down(D) regulated in PTC with respect to NOR or/and significantly synergistically (S)/antagonistically (A)/independently (I) expressed with *SPINT2* in NOR or/and PTC.

GENE	DESCRIPTION	NOR	PTC	REG
*ACTG1*	actin, γ 1	S		
*AKT1*	v-akt murine thymoma viral oncogene homolog 1			U
*AKT3*	v-akt murine thymoma viral oncogene homolog 3		S	
*ATF4*	activating transcription factor 4	A		
*ATM*	ataxia telangiectasia mutated		S	
*BAK1*	BCL2-antagonist/killer 1		S	
*BAX*	BCL2-associated X protein			U
*BBC3*	BCL2 binding component 3		S	U
BCL2	B-cell CLL/lymphoma 2	I		
*BCL2A1*	BCL2-related protein A1			U
*BCL2L1*	BCL2-like 1		S	
*BCL2L11*	BCL2-like 11		S	
*BID*	BH3 interacting domain death agonist			U
*BIRC5*	baculoviral IAP repeat containing 5			U
*CAPN1*	calpain 1, (mu/I) large subunit		S	
*CASP2*	caspase 2, apoptosis-related cysteine peptidase		S	
*CASP3*	caspase 3, apoptosis-related cysteine peptidase			U
*CASP6*	caspase 6, apoptosis-related cysteine peptidase		S	
*CASP9*	caspase 9, apoptosis-related cysteine peptidase	S		
*CFLAR*	CASP8 and FADD-like apoptosis regulator		S	
*CTSC*	cathepsin C			U
*CTSD*	cathepsin D		S	
*CTSH*	cathepsin H			U
*CTSK*	cathepsin K			D
*CTSL*	cathepsin L	S		
*CTSV*	cathepsin V		S	
*CYCS*	cytochrome c, somatic	S		D
*DAB2IP*	DAB2 interacting protein		S	U
*DFFA*	DNA fragmentation factor		S	
*DIABLO*	diablo, IAP-binding mitochondrial protein		S	U
*EIF2AK3*	eukaryotic translation initiation factor 2-α kinase 3		S	
*EIF2S1*	eukaryotic translation initiation factor 2, subunit 1 α		S	
*FAS*	Fas cell surface death receptor			U
*FOS*	FBJ murine osteosarcoma viral oncogene homolog			D
*GZMB*	granzyme B (granzyme 2, cytotoxic T-lymphocyte-associated serine esterase 1)		S	D
*IKBKB*	inhibitor of kappa light polypeptide gene enhancer in B-cells, kinase β	I	S	
*ITPR2*	inositol 1,4,5-trisphosphate receptor, type 2			U
*JUN*	jun proto-oncogene			D
*LMNA*	lamin A/C			U
*LMNB2*	lamin B2		S	
*MAP2K1*	mitogen-activated protein kinase kinase 1			U
*MAP2K2*	mitogen-activated protein kinase kinase 2		S	
*MAP3K9*	mitogen-activated protein kinase kinase kinase 9		S	
*MAPK1*	mitogen-activated protein kinase 1		S	
*MAPK3*	mitogen-activated protein kinase 3		S	
*NFKB1*	nuclear factor of kappa light polypeptide gene enhancer in B-cells 1	S		
*NFKBIA*	nuclear factor of kappa light polypeptide gene enhancer in B-cells inhibitor, α			D
*NRAS*	neuroblastoma RAS viral (v-ras) oncogene homolog		S	
*PARP1*	poly (ADP-ribose) polymerase 1			U
*PARP4*	poly (ADP-ribose) polymerase family, member 4		S	
*PDPK1*	3-phosphoinositide dependent protein kinase 1		S	
*PIK3CA*	phosphatidylinositol-4,5-bisphosphate 3-kinase, catalytic subunit α		S	
*PIK3R1*	phosphoinositide-3-kinase, regulatory subunit 1			D
*PIK3R2*	phosphoinositide-3-kinase, regulatory subunit 2		S	U
*PMAIP1*	phorbol-12-myristate-13-acetate-induced protein 1		S	U
*RAF1*	v-raf-1 murine leukemia viral oncogene homolog 1	S		
*RELA*	v-rel avian reticuloendotheliosis viral oncogene homolog A		S	
*TNFRSF10B*	tumor necrosis factor receptor superfamily, member 10b			U
*TNFRSF1A*	tumor necrosis factor receptor superfamily, member 1A		S	U
*TRAF2*	TNF receptor-associated factor 2		S	
TUBA1C	tubulin, α 1b	I		
TUBA3D	tubulin, α 3c	I		
*TUBA4A*	tubulin, α 4a			U
*XIAP*	X-linked inhibitor of apoptosis		S

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
