# Peer review of "Biomarkers, Master Regulators and Genomic Fabric Remodeling in a Case of Papillary Thyroid Carcinoma"

_genes, 2020, doi:10.3390/genes11091030_

Round 1

Reviewer 1 Report

In the manuscript by DA Iacobas entitled, “Biomarkers, master regulators and genomic fabric remodeling”, the author presents a novel prognostic algorithm to identify transcriptome synergism and key molecular pathway regulators using a patient thyroid cancer sample compared to control tissue from the same individual. Validation of these bioinformatic predictive approaches were performed using transcriptomes of genetically manipulated papillary (BCPAP) and anaplastic (8505C) human thyroid cancer cell lines distinct from the patient sample used in the present study. The use of expression coordination calculation to define in-phase, anti-phase, and independent expression dynamics is an innovative metric as it is presented in the present paper, however the conclusions made regarding biological functional changes in the absence of biochemical/genetic/molecular assay data detracts from the overall strength of the paper. The author may wish to consider revising the language to reflect the bioinformatic analytical focus of the manuscript, or provide additional experimental support for one or more of the assertions made about changes in thyroid function, cell cycle dynamics, mitochondrial function (oxidative phosphorylation), and SPINT2 gene regulatory function.

The following specific comments/concerns should be addressed by the author:

Comments:

  • Syntax errors have been noted all throughout the submission. These should be addressed to improve overall readability of the manuscript.
  • CHI3L1 and TFF3 were the most extremely dysregulated genes, does that become irrelevant when calculating expression coordination?
  • What is the rationale for the focus on apoptosis, cell cycle and oxphos pathways over others?
  • While the rationale for using DDX19B for the validation of this algorithm can be understood, the way the manuscript is currently written is slightly confusing as it gives the impression that SPINT2 and DDX19B are functionally related in the context of thyroid cancer.
  • As I understand it, SPINT2 has high expression stability and gene coordination that mark it as a GMR according to the present bioinformatic analysis. While the focus in the present study was in the context of apoptosis genes, how does SPINT2 regulate genes in other pathways? Is its coordination only high with apoptosis genes?
  • Duplication of Refs #18 and #25
  • Figure 1 – inconsistent formatting between x-axis labels in (b) and (c)
  • Figure 3 – While the legend has been provided, the up-regulated genes are not legible when printed. Consider improving contrast to make text readable. Also, the legend is embedded in a very busy figure – perhaps a simple legend located in the corner would be a more efficient layout.
  • Figure 4 – Please seem comments for Figure 3.
  • Figure 5 – Differential orientation of figure panel designation and text within the figure make this an awkward read. Also there is no legend included with the text.
  • Figure 7 – As a suggestion, perhaps this information might be best served if presented as a table.

Author Response

Thank you for revising our work and very useful suggestions. Please see the attachment.

Reviewer 2 Report

In this manuscript the author re-analyze previously generated transcriptomic data obtained from one case of papillary thyroid carcinoma (PTC) and from genetically manipulated BCPAP and 8505C human thyroid cancer cell lines.

The performed analysis uses a method by previously published and validated by the same author (ref. 7 to 9).

In the previous publications the concept of Gene Commanding Height (GCH) was introduced as a composite measure of gene expression control and coordination that is indicative of how important is the expression of a gene for the phenotype of a cell.

This measure is then used to identify Gene Master Regulators (GMRs).

As of my understanding the main hypothesis behind this and previous publications from the same author is that cancer and normal cells are governed by distinct gene master regulators (GMRs).

The author also explains how these GMRs are different from biomarkers and why the therapeutical targeting of the first rather then the latter would lead to a better outcome.

In the present manuscript  the performed analysis identified the serine peptidase inhibitor, Kunitz type, 2 (SPINT2) as a GMR (in the analyzed PTC case) . 

The author then shows how, according to his analysis, overexpression of SPINT2 would be effective in inducing programmed cell death in PTC cells by upreregulating apopotosis genes.

I believe the manuscript is generally well written and the figures are clear enough.

I do suggest improving the quality of the text in figure 3 and 5 as some of it is a bit difficult to read.

Also the legend of figures 8 and 6 are confusing since it looks like the caption in bold takes most part of the figure legend.

The author should also elaborate more in the conclusion part.

Author Response

Thank you for the review and useful suggestions. Please see the atachment.

Reviewer 3 Report

Dear author,

thank you very much for an interesting article concerning the alteration of functional pathways in papillary thyroid cancer. Available transcriptomic data were analyzed and the authors established the gene hierarchy, identify potential gene targets, and predict the effects of their manipulation. They identified serine peptidase inhibitor, Kunitz type, 2 (SPINT2) as the Gene Master regulator of PTC. In addition, they found SPINT2 to be a relevant factor in tumor development/progression.

The manuscript is well written - especially your results section is well written including several high-quality images and I recommend after minor revision.

1) Please emphasize the identified biomarkers of your study at the beginning of your discussion. Please write detailed.

2) Please include a limitation section in the discussion e.g. page 17 l. 468: TP53 is one of the most interesting cancer-related genes and unfortunately not quantified in the experiment

3) Your conclusion is too vague, please rewrite your conclusion e.g. at least 3-5 sentences

Author Response

Thank you very much for the kind appreciation and useful recommendation. Please see the attachment.

Round 2

Reviewer 1 Report

Thank you for addressing all previous comments.